# Turfgrass Disease Diagnosis: Past, Present, and Future

**DOI:** 10.3390/plants9111544

**Published:** 2020-11-11

**Authors:** Tammy Stackhouse, Alfredo D. Martinez-Espinoza, Md Emran Ali

**Affiliations:** 1Department of Plant Pathology, University of Georgia, Tifton, GA 31793, USA; tstackho@uga.edu; 2Department of Plant Pathology, University of Georgia, Griffin, GA 30223, USA; amartine@uga.edu

**Keywords:** plant pathology, disease detection, biotechnology, history, molecular detection, disease monitoring

## Abstract

Turfgrass is a multibillion-dollar industry severely affected by plant pathogens including fungi, bacteria, viruses, and nematodes. Many of the diseases in turfgrass have similar signs and symptoms, making it difficult to diagnose the specific problem pathogen. Incorrect diagnosis leads to the delay of treatment and excessive use of chemicals. To effectively control these diseases, it is important to have rapid and accurate detection systems in the early stages of infection that harbor relatively low pathogen populations. There are many methods for diagnosing pathogens on turfgrass. Traditional methods include symptoms, morphology, and microscopy identification. These have been followed by nucleic acid detection and onsite detection techniques. Many of these methods allow for rapid diagnosis, some even within the field without much expertise. There are several methods that have great potential, such as high-throughput sequencing and remote sensing. Utilization of these techniques for disease diagnosis allows for faster and accurate disease diagnosis and a reduction in damage and cost of control. Understanding of each of these techniques can allow researchers to select which method is best suited for their pathogen of interest. The objective of this article is to provide an overview of the turfgrass diagnostics efforts used and highlight prospects for disease detection.

## 1. Introduction

Turfgrass is a multibillion-dollar industry that encompasses lawns, parks, sports fields, and golf courses with over 62 million acres in the US alone [1]. A study in 2006 [2] found that across the US, the turfgrass industry generated over 58 billion dollars (83 billion with inflation from 2002 to 2020) annually [3]. Turfgrass has many positive environmental impacts, including reducing temperatures in a given area [4], energy use reduction [5,6], potential phytoremediation uses [7,8,9], and erosion control [10,11]. It has also been shown that greenscapes such as turf can reduce stress and increase cognitive capabilities [12,13]. These benefits contribute to why turf is so widely grown, with rapid increase in use around the world [1]. While there are some concerns about potential negatives of turfgrass growth, including water use and chemical use, the positive impacts outweigh the negatives and these disadvantages can be reduced with breeding drought and disease tolerant varieties [14].

While the turfgrass industry is rapidly growing, there are issues that make turf management more difficult, including losses from abiotic (non-living) and biotic (living) factors. Abiotic factors include extreme moisture, extreme temperature, excess or deficient water, nutrient deficiencies, chemical damage, mechanical damage, and/or adverse cultural practices. These stresses can cause damage to turfgrass and can easily be confused with biotic disease problems [15]. If misdiagnosed, the turfgrass manager may spray costly chemicals that will not remedy the problem and instead delay the time until proper management. Biotic agents are various types of plant pathogens that can affect turfgrass health. Most turfgrass diseases are caused by fungi; however other groups cause damage, including bacteria, viruses, and nematodes. The cost of damage varies depending on the turfgrass use and extent of management. In Georgia during 2017 alone the cost of losses due to damage and control for soilborne, foliar, crown, and nematode diseases equaled around 150 million dollars [16]. It was also approximated that these diseases cause a 5% reduction in crop value [16]. Pathology diagnosis research in this industry has been rapidly expanding (Figure 1) with the goal being rapid detection to reduce pathogen control costs.

There are several factors that make diagnosing diseases in turfgrass difficult. Plants grown as “turfgrass” encompass about 50 different species of grasses, each with their own varieties and growth requirements. Pathogenic bacteria, fungi, nematodes, and viruses cause a myriad of symptoms on the grasses. Some pathogens will cause different symptoms depending on the grass species and certain signs and symptoms from different pathogens can appear similar, making diagnosis difficult. Additionally, there can be several pathogenic species causing a disease with the same name, allowing for confusion when referring to fungicide resistances. Treatment can be a challenge without proper identification, as different pathogens and even strains of the same species respond differently to various chemical treatments. While disease tolerance can be relatively easy to breed for in other crops [17,18], turfgrass disease resistance has proven difficult to accomplish [19]. In addition, many pathogens are difficult or impossible to culture, making traditional disease screening extremely challenging and sometimes not possible, as with viruses.

It is important to screen plants early into an infection, with non-symptomatic predictions being the ideal method in preventing widespread damage in low damage threshold locations. A rapid response for pathogen diagnosis is vital, as generally the damage caused by these pathogens can be quick to appear and spread, but slow to recover [20]. While the presence of a pathogen itself does not denote disease, in some uses turfgrass has a particularly low threshold for damage, particularly in sports fields that must be uniform [21]. In these locations simply the presence of the pathogen on a susceptible host can be of great concern, even if the environmental conditions required for disease are not yet met. In sports fields specifically, screening for known pathogenic organisms may allow for disease prevention. Furthermore, the identification of the early stage of infection is important to take the appropriate swift measures. This can prevent the further spread of disease as a result in cost savings and a reduction in loss due to diseases.

The purpose of this review is to summarize the status of pathogen detection within the turfgrass field (Figure 1). This review focuses on pathogen identification and the various methods used for detection. While there are examples for all pathogens with available assays, the hosts are not covered to keep the focus on a universally helpful diagnosis review. Specific pathogen examples are meant to cover what is available and not giving preference to any areas or hosts. This review also does not cover control measures. In the sections below, we first provide an overview of the traditional disease detection methods like signs, symptoms, morphology, and ELISA testing. Next, nucleic acid-based techniques including various PCR and sequencing techniques are examined, followed by portable isothermal reactions like LAMP and RPA methods. Finally, certain future methods being utilized in other plant pathology areas are detailed, with potential uses within turfgrass. Each diagnosis method comes with its own set of advantages and limitations; therefore, each different method of diagnosis should be weighed before developing new tools for diagnostics. This review will allow for a comprehensive understanding of the current methods in use within the turfgrass industry for pathogen detection, and the individual benefits and drawbacks of each method.
Figure 1A brief history of pathogen detection within turfgrass. The figure states an approximate starting date for pathogen diagnosis using signs and symptoms [22,23], plate culturing [22], compound microscopy [24], advanced microscopy [25], ELISA [26], lateral flow devices [27], PCR [28,29], qPCR [30], multiplex PCR [31], genome sequencing [32], loop-mediated amplification [33], recombinase polymerase amplification [34], portable nanopore sequencing [35], advanced remote sensing [36], and disease management via phone applications [37].
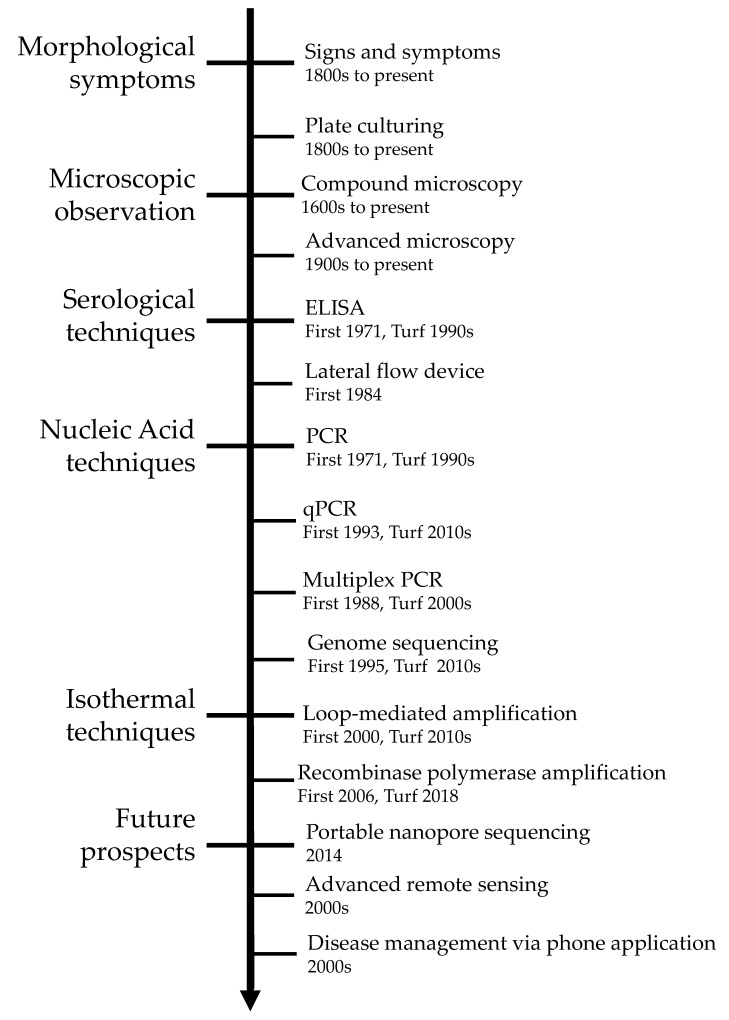



## 2. Traditional Disease Detection Methods

Conventionally, the diagnosis of turf grass pathogens relies on visible disease signs and symptoms and microbial cultures with microscopy. ELISA assays are an immunosorbent assay that has been widely used for disease diagnosis since the 1970s. These methods are widely used and have been established for decades.

### 2.1. Pathogen Signs and Disease Symptoms

The first and arguably the most important part of pathogen identification is determining the pathogen signs and disease symptoms. Disease diagnosis typically goes through a series of steps. First the disease symptoms are described, and the host and environment are noted (Figure 2). The symptoms for a given disease may vary depending on the host or environment, so host identification is vital when trying to identify a disease. Next, samples of the potentially infected host are taken. For turfgrass diagnosis it is common to collect the entire plant ecosystem using an 11 cm plug [21]. The sample is closely examined for signs of pathogens and can be used to reach a preliminary diagnosis to help decide which confirmation test to utilize. The potential pathogens are then identified via the selected detection and confirmation methods. Note that the presence of a pathogen alone does not denote the pathogen is causing symptoms; however, having a known pathogen on a host with symptoms previously shown to be caused by that pathogen is enough cause for treatment. Moving forward, it is also vital to examine how the environmental conditions can be changed to reduce or prevent disease pressure in the future. There are many excellent resources available for turfgrass disease identification using traditional methods of sign and symptom diagnosis [21,38,39,40,41]. Based solely on disease symptoms, however, determining whether the condition is caused by a biotic or an abiotic agent can be challenging, due to similar symptoms. There are many abiotic stressors that can appear similar to disease symptoms [15]. It requires expertise that takes extensive training and years of practice to perfect and prevent incorrect diagnosis. This strategy is typically not very helpful in early disease infection stages because damage is required for diagnosis.

### 2.2. Microscopy and Culturing

Traditionally, microscopy has been used to identify most plant pathogens, including turfgrass pathogens. For pathogen identification via microscopy, a pathogen sample is taken, isolated, and viewed under a compound microscope (Figure 3). While this method requires very simple equipment, it requires specially trained personnel who can identify the pathogen. There are also limitations with microscopy, especially distinguishing exact species [42]. Some species also have several related species with similar host symptoms or species physiology that make species identification difficult, such as *Gaeumannomyces graminis* the causal agent of the take-all disease [43]. Some pathogens, like viruses, are nonculturable and too small for microscopy, preventing this method from being useful for those infections. The expansion of molecular tools have caused the names of some pathogen change over time, such as with *Clarireedia* spp., the causal agent of dollar spot [44]. This can cause confusion with using outdated microscopy keys. Even still, microscopy is a widely used, very useful, reliable, and cost effective tool for preliminary diagnosis in turfgrass [45], and in some cases it is still the best method [40,46,47,48]. The use of molecular tools allows a diagnostician to either bypass microscopy or confirm preliminary results, depending on the strategy used.

## 3. Serological Techniques

Techniques that are specific to a given pathogen can be extremely helpful in identification. There are several serological techniques based on antibody detection used within plant pathology disease diagnosis. These include enzyme-linked immunosorbent assays (ELISAs) and immunostrip testing.

### 3.1. Enzyme-Linked Immunosorbent Assay (ELISA)

One of the first methods invented for testing for a specific pathogen in a sample is the enzyme-linked immunosorbent assay (ELISA) [26]. Double antibody sandwich ELISA (DAS-ELISA) is the most commonly used because it is more sensitive than other ELISA methods and allows for quantification with a plate reader and validated controls, though is still more often used for qualitative testing with visual interpretation [49]. This method uses antibodies to detect antigens within a mixture of proteins from the diseased tissue. If the specific antigen that matches the antibody is present in a sample the antigen will bind to the antibody. A visual color change or a quantitative machine like a plate reader allows the user to confirm the sample as positive or negative (Figure 4). The disadvantages of the ELISA methods are that the sensitivity is not as high as many other molecular methods and it takes a long time (24 h) to perform the assay. With the fast spreading pathogens typical to turfgrass, that amount of time can be method prohibitive. The sensitivity limitations make this method most useful when the symptoms have had time to develop, which turf managers would like to avoid. A major advantage of ELISA is, if testing many samples, ELISA testing is extremely cost effective. ELISA methods are available to the turf industry (Table 1) to detect *Pythium* spp., *Leptosphaeria*, and *Rhizoctonia* [50,51,52,53]; however, their use is not particularly common. There is use of sugarcane mosaic virus (SCMV) ELISA tests in some areas, but the low sensitivity and high time commitments keep this method from further widespread use.

### 3.2. Lateral Flow Device

Lateral flow device assays allow for simple disease diagnosis in the field with little to no expertise on a given pathogen. The earliest use of the term “lateral flow device” was in the early 70s in medical diagnosis [74], and the concept has not changed. Lateral flow devices work similarly to an ELISA test, with specific antibodies diffusing when in contact with a specific antigen. In the case of lateral flow devices, capillary action carries the antibodies into contact with the antisera at a specific location. When a positive sample containing the antibodies meets the antisera, there will be a mark, called a band, to denote that the sample is positive for a given antigen (Figure 4). These tests also include a positive control, a second band that simply tests if the antibodies were moved through capillary action to ensure a functional assay. The assay can be used in fields with results in under an hour. This technology has been used for decades in medicine, with the most well-known lateral flow device assay being a pregnancy test but has been commercially available in plant pathology from companies like Agdia since the early 2000s. In plant disease diagnosis, lateral flow devices are most used for virus and bacterial diseases such as *Ralstonia solanacearum* [75] but also has been used for oomycetes like *Phytophthora* [76]. These assays have not yet been used for turfgrass disease diagnosis, but have great potential due to their speed, portability, and ease of use. Currently, the biggest drawback of lateral flow devices is the limited number of available assays and the difficulty in developing new ones with high specificity.

## 4. Nucleic Acid-Based Techniques

With the rapid development of molecular biology, the diagnosis technology of plant pathogens has developed from the traditional morphological diagnosis to the current molecular diagnosis. The molecular detection techniques of pathogens include conventional polymerase chain reaction (PCR), multiplex PCR, quantitative PCR (qPCR), RAPD-PCR, and DNA sequencing methods.

### 4.1. Conventional Polymerase Chain Reaction (PCR)

Polymerase chain reaction (PCR) is an extremely common and effective method of identifying a pathogen sample. It works by using two primers, short nucleic acid sequences that complement a specific sequence within a sample, along with dNTPs, polymerase, and other components to amplify that specific sequence. The result is an exponential copying of a specific region within the sample tube. This small fragment, specific size varies from one assay to another, can be viewed on a gel using gel electrophoresis (Figure 5A). This allows a PCR product fragment to be seen, and the fragment’s size to be roughly determined to compare to the expected fragment size. If the target gene sequence was present, there will be a specific sized band present. If the target sequence was absent, the band will not be present, or there will be a band at the wrong size. This entire process takes 2–4 h and some basic expertise to run the test. The target region can vary; conventional PCR identifies pathogens by designing specific primers that target the specific gene for exponential amplification of a sequence. This method does require having a preliminary diagnosis but allows for confirmation on a molecular level within a few hours. This method relies on having short sequences of DNA that are unique to that specific pathogen. When these primers are designed, they are typically tested against closely related species to ensure specificity, and in silica testing against sequences in the online database GenBank. Genus or species-specific PCR has been used in turfgrass disease diagnosis for over two decades (Table 1) [43,54,55,56,57,58,59,60,61,62]. Genus specific primers have the drawback that if another closely related, nonpathogenic species is present then the sample will be positive, even if that pathogen is not causing disease. Species-specific primers avoid this problem but can be harder to design. A major disadvantage of this methodology is that if the initial diagnosis is incorrect then a re-evaluation is needed, and a different method may be required. This means the method still requires expertise to make an initial diagnosis.

### 4.2. Multiplex Polymerase Chain Reaction

Multiplex PCR is a PCR that can be adapted to allow for several different tests to occur at one time in one reaction [31]. This is accomplished using species-specific primers that have similar temperature requirements and produce different sized bands visualized with gel electrophoresis (Figure 5B). Each primer set has a unique band size and multiplexing allows for one sample to be tested for several pathogens simultaneously. Each different band size is read with the ladder to determine whether a pathogen or pathogens are present from the species being tested. Multiplex PCR speeds up the testing time compared to running several assays, with multiplex PCR taking the same 2–4 h seen in conventional PCR. It also reduces the cost of materials as only one reaction is needed. The major disadvantage is that not many of these assays have been designed and band sizes can be difficult if designed too close together. Some equipment can make this portion easier, such as the QIAxcel Advanced capillary electrophoresis system (Germantown, MD). Special care needs to be taken when producing primers and multiplex PCR may not always be an option; however, this method has already shown its usefulness in turfgrass to detect various *Pythium* spp. and *Gaeumannomyces graminis* [54,56]. This method can also be used in conjunction with other methods, such as quantitative PCR [78].

### 4.3. Quantitative Polymerase Chain Reaction (qPCR)

Quantitative PCR (qPCR) is a modified PCR protocol that allows for extremely sensitive, real time, and quantitative analysis of PCR reactions [79,80]. qPCR uses a different polymerase than conventional PCR, plus has an addition of either an intercalating tracking dye, like SYBR Green, or an assay specific probe, like a TaqMan probe or molecular beacon [30]. This modified PCR protocol requires a different thermocycler than conventional PCR; this machine will have fluorescent detection capabilities. This change allows for a quantification of samples and increases sensitivity (Figure 5C). How qPCR quantitation works depends on the marker type used. SYBR Green works by intercalating within DNA with a fluorescent signal. Probes work similarly, with their signal becoming florescent as the probe is cleaved (TaqMan) or hybridized (molecular beacons). As more DNA fragments are exponentially produced in PCR, more of the signal can be read, allowing for real time quantification. qPCR is often seen as the most sensitive detection method in disease diagnosis. It is also rapid, as the qPCR process takes about 1 h after DNA or RNA extraction. It also does not require a gel step, like PCR and multiplex PCR. It is often used in clean stock seed certifications or human pathogen detection. qPCR is extremely useful, sensitive, and has widespread popularity, but requires expensive equipment and highly skilled personnel. qPCR has already been used to diagnose certain turfgrass pathogens including *Clarireedia* sp., *Magnaporthe poae*, *Rhizoctonia solani*, and *Puccinia coronata* (Table 1) [63,64,65,66,67].

### 4.4. Random Amplification of Polymorphic DNA PCR

Random amplification of polymorphic DNA PCR (RAPD-PCR) is a method to read segments of DNA for a profile of a species without knowledge of the species beforehand. It works by using a short random primer to amplify a random sequence, then the products are run on a gel to separate the products by size. The bands are extracted, cloned, and sequenced to characterize that area. The amount of work required and difficulty in reproducibility made this method difficult. This method is no longer used, but was utilized in plant disease diagnosis and turfgrass fields to create species-specific assays before other methods were discovered [57,81,82,83].

### 4.5. DNA Sequencing

Sanger DNA sequencing is used to ID plant pathogens. Universal primers can amplify many organisms with identification performed with sequencing. There are some genes or sequences that are highly conserved, so those specific portions of DNA are unlikely to change over time or between species. These sequences can be targeted with generic primers to allow for the less conserved sequences between the two primers to be amplified and read via Sanger sequencing. The sequencing tags bases with a florescent protein and reads each specific base pair (Figure 6). This method allows for a pathogen sample to be identified with relatively little information, such as knowing the sample is a bacterium, its family of viruses, true fungi, or oomycete. There are many “universal primers” for a kingdom of pathogens. For bacterial pathogens, the 16s rRNA [84], β- operon protein gene, or outer membrane genes are typically targeted. For fungal pathogens, the internal transcribed spacer (ITS) region is typically used for identification [85]. For viral pathogens, RNA based viruses will require an additional step of the reverse transcription, then typical universal targets include capsid protein (CP), three prime untranslated regions (3′ UTR), and replicase protein [86]. The resulting sequence is compared to other previously sequenced fragments in the online database GenBank. This method is also extremely helpful with emerging pathogens that may not have been considered if it has not been reported within a given area [87,88]. While universal primers amplify across a wide range of potential pathogens requiring for little identification knowledge, sequencing often requires a much longer time to get results over other methods, typically a few days [89]. Beyond the time difference, this method often requires having a pure culture of the pathogen, a requirement that can be extremely difficult to achieve, if possible at all. The method will not work if several organisms are amplified and special care is taken in selecting the proper universal primer. Another weakness is that sequencing can fail through failed runs, weak chromatograms, unclean amplicons, and other problems, requiring several more days of processing for a second result. Additionally, if a pathogen sequence is not present in GenBank samples cannot be identified with this method.

### 4.6. Next Generation Sequencing

Next generation sequencing (NGS) allows for rapid sequencing and identification of pathogens through complete genome sequencing [90]. Next generation sequencing is used for drafting genomes of pathogens or to discover single nucleotide polymorphisms (SNPs) and single nucleotide variants (SNVs) in a sample [91,92,93,94,95]. This information can be used to determine the pathogen race or lineage [93]. This information can be used for tracing the spread of pathogens and understanding characteristics like fungicide resistance. Some NGS technology can be used to directly identify the samples without culture, including those that cannot be cultured and cannot be identified by other technologies [96]. Next generation sequencing is still much more expensive than other methods and can take a long time to process a sample and the resulting data; however, the cost and time needed are decreasing rapidly over time. With decreasing cost and increasing accessibility of analytical platforms, NGS and genome-based species identification and detection could revolutionize the diagnosis of oomycete pathogens such as *Pythium* and *Phytophthora* spp. [97]. This technology has been utilized in turfgrass for full genome sequencing of *Clarireedia* spp. the dollar spot causing pathogens [95]. The development and application of these and other technologies will provide more efficient detection of turfgrass pathogens.

## 5. Isothermal Diagnosis Techniques

Rapid disease diagnosis is vital in responding to a disease outbreak early. To this end, there are several techniques that allow for disease diagnosis with an isothermal reaction. These allow for fast treatment without a thermocycler and laboratory setting. There are many isothermal techniques, with the most used currently in turfgrass and plant pathology being loop-mediated amplification (LAMP) and recombinase polymerase amplification (RPA).

### 5.1. Loop-Mediated Amplification (LAMP)

A recent development in molecular disease testing is loop-mediated amplification (LAMP). This method works by using four to six uniquely designed primers to target six to eight regions within a target sequence [33,98]. This process forms various sized concatemers of DNA from a sample of DNA or RNA that allow for exponential amplification [33]. This method can be performed in under an hour with isothermal conditions, removing the need for a thermocycler and allowing for in-field testing. Results can be read via several methods, including a color change [99], turbidity test [100], gel electrophoresis (Figure 7A) [33], or quantitative fluorescence-sensing piece of equipment (Figure 6B), such as a Genie [101]. The specificity of LAMP is already high due to the large amount of sequence covered by the primers, but probes can also be used to improve specificity. The sensitivity of LAMP assays is much higher than that of conventional PCR and allows for earlier detection of pathogens [102,103]. This method has been used throughout plant pathology. The turfgrass industry already has many (some) LAMP assays for pathogen identification with some notables including: *Gaeumannomyces avenae*, *Ophiosphaerella korrae*, *Magnaporthiopsis poael*, *Rhizoctonia solani*, and *Xanthomonas translucens* (Table 1) [68,69,70,71,72,73]. While these assays exist for turfgrass pathogens, it is unclear how widely they are currently being used for diagnosis. Despite its high effectiveness, the cross contamination is the biggest challenge of LAMP and usually leads to false-positive results. Another limitation of this method is the difficulty in designing primers, as they require 4–6 regions in the sequence. Further improvements are needed to use this technology for robust turf grass disease diagnosis.

### 5.2. Recombinase Polymerase Amplification (RPA)

Recombinase polymerase amplification (RPA) is another isothermal molecular assay that can be used for molecular detection. RPA can be run anywhere with a under 20-min reaction time and temperature requirements of 22–45 °C [104,105]. RPA requires two primer sets, like a conventional PCR reaction, but has some added reagents that allow for an isothermal reaction. First, RPA includes a recombinase enzyme that allows the primers to bind within double-stranded DNA at their homologous sequence, displacing the strands. Then a single-stranded DNA binding (SSB) protein can attach to the displaced DNA, stabilizing the structure. Then a polymerase attaches and begins to replicate the strand, much like conventional PCR, but the polymerase is unique with strand displacing qualities. DNA or cDNA can be detected with RPA, and reverse transcriptase can be added to allow for RNA detection [105]. If the sequence is present, there will be exponential amplification of the target sequence; however, without the target sequence amplification will not occur. The results can be read with gel electrophoresis [105], a real-time amplification machine like a plate reader (Figure 7B) [34], or an immunostrip (Figure 7B) [34]. RPA has been reported to be extremely sensitive, with some studies stating a modified RPA method can detect as low as 1–10 copies and 100-fold more sensitive than qPCR [104,106,107]. RPA has been reported to be highly specific; however, it has been shown even with many mismatched primer nucleotides there have been reports of amplification with RPA [108,109]. This leads to concerns about non-specificity, particularly in the presence of closely related species [104]. To our knowledge, this method has only been used for turfgrass pathogen detection with *Gaeumannomyces avenae*, *Ophiosphaerella korrae*, and *Magnaporthiopsis poae* (Table 1) [68]. It has been used with many other plant pathogens, including viruses (little cherry virus 2, plum pox virus, and tomato chlorotic dwarf viroid) [110,111,112,113], bacteria (*Candidatus Liberibacter asiaticus*) [114], and fungi (*Fusarium oxysporum*) [115].

## 6. Future Outlooks in Diagnostic Identification

Often it is difficult to predict the direction a field will advance moving forward. There are many techniques for disease diagnosis that are being developed in plant pathology right now. We have included a brief introduction to several techniques that could be extremely useful in turfgrass disease detection including: portable high-throughput sequencing, remote sensing, and smartphone applications for disease tracking.

### 6.1. Portable High-Throughput Sequencing

High throughput sequencing is moving to portable technology, allowing for on the go sequencing solutions. Oxford Nanopore Technology (ONT) is a type of high throughput sequencing that uses tools like the portable, phone-sized MinION genome sequencer that allows for rapid sequencing of samples with up to 100 kb reads without cloning in any setting, including field diagnosis [116]. ONT works by pulling individual strands of DNA or RNA through a nanopore that reads each nucleotide by measuring changes in electrical current. This technology allows mixed samples of unknown components to be read, allowing for identification of pathogens without preliminary diagnosis [117]. It is extremely useful in circumstances where there may be several pathogens infecting at once, but there may be many non-pathogenic species present as well, making the data harder to sort and requiring extensive training. Sorting through the massive amount of data produced is not fast or easy. It requires a highly trained user that has the time to sort the information and an understanding of what organisms will be present without causing disease. This method is also extremely expensive, costing about ten-fold more than any other method discussed in this review. Portable high-throughput sequencing technology has already launched for onsite disease detection for plants and humans [118,119,120]. It has not yet been used in turfgrass.

### 6.2. Remote Sensing of Pathogens

Another technology that could be incredibly useful for disease detection in turfgrass is remote sensing of stress and disease through hyperspectral imaging. The beginnings of this field were back in the 1980s, but have been fast adapting over the last decade [121]. This technology is based on detecting spectral differences and reflection sensitivity of both healthy and diseased tissue, specifically looking at visible (VIS, 400–700 nm), near-infrared (NIR, 700–1000 nm), and shortwave infrared wavelengths (SWIR, 1000–2500 nm) light ranges [122,123]. Most studies in this field collect data using handheld spectroradiometers, but some studies show the promise of airborne sensor use [124,125], similar to previous studies using hands-off satellite or photographic imaging to study vegetative changes over time [126,127,128,129,130]. In some studies, the disease can be sensed before it is visible to the naked eye and different diseases give off different signatures [131,132,133]. The data can also give information on abiotic issues, like water or nutrient stress [134]. The data is also moving closer to automation, with less human input needed in data analysis [135]. Thus far, the technology has been used in many plant studies, including: rice [124], tomatoes [136,137], sugar beet [132,133], barley and wheat [131,135] and potatoes [138,139]. This could be a vital tool for catching a disease before it becomes a problem. The biggest issues with this method right now are how few crops have been studied for this method and the initial cost of setting up this kind of disease sensing program. These issues will have to be addressed before advanced remote sensing is widely used or introduced in turfgrass programs.

### 6.3. Smartphone Applications for Detection and Management

Smart phone applications and online programs can be used to make disease prediction and diagnosis easier for growers. There are many smartphone applications created to diagnose plant diseases based on photographs, keep track of chemical treatment schedules, help manage maintenance, and other useful features for growers (Figure 8) [140,141,142]. These applications can have information added and edited as science progresses. There are also artificial intelligence programs being developed to better these applications ability to give information [141]. The ability to predict disease outbreaks is an important goal in disease surveillance and pathogen detection. Some of these programs will use weather information to better predict a pathogen outbreak [143,144]. This can allow for early spraying before the damage has occurred and a reduction in unneeded chemical use. The biggest downside to this technology is that the prediction information is not available everywhere, due to the extensive use of weather data. Additionally, these applications must be updated and maintained, but the ability to update as new information develops is a benefit for this format of information dissemination. A drawback for diagnosis applications is the AI may have trouble distinguishing diseases with similar symptoms. Some management applications exist for turfgrass (Figure 8), but some of the more advanced features, like weather tracking for disease prediction and AI disease identification, have yet to be developed for specifically turfgrass [37]. However, there have been applications available for management, chemical treatments, and pathogen information since 2005, with many options on the market today.

## 7. Conclusions

At present, turfgrass pathogen diagnostics mainly rely on traditional microcopy and PCR-based technologies. Some pathogens have LAMP and RPA diagnosis, but those methods are certainly not all inclusive. In general, for known and culturable pathogens, microscopy, and PCR amplification using universal ITS primers are the most used assays for detection of turfgrass pathogens, which can take several days for diagnosis. For unknown and unculturable pathogens there are very few methods that can be used. Outside of dollar spot disease, next generation sequencing has not been performed in turfgrass, making further genetic studies difficult.

There are many advantages and disadvantages (Table 2) for each diagnosis method that have been or will likely be used in turfgrass. While earlier methods like culturing and morphology can be quick, it requires technical expertise and can be difficult or impossible with some species. Nucleic assay techniques can take longer than the morphology diagnosis, but they have precision and sensitivity that earlier methods can lack. Onsite techniques allow for extremely quick action and do not require waiting for a sample to arrive at a laboratory. They can be extremely sensitive, but their specificity can vary, and some methods (lateral flow devices) have limited assays as of this writing. Future methods will likely include NGS studies that can be used to study favorable traits such as tolerance and remote sensing, which has been shown to detect some pathogens before damage is visible to the naked eye. These methods taken together allow for researchers to build the tools needed to protect the assets of the turfgrass industry.

Species differentiation is important to track the movement of each species and allow for further work, such as genetic studies, fungicide resistant assays, and species range. Methods like universal primer sequencing and next generation sequencing can be used to study the pathogen or host genome for specific solutions to disease attacks or for chemical resistance within the pathogen. This can allow for genetic modification or selective breeding to better protect the grass.

Molecular methods allow for a faster response to disease outbreaks and therefore reduced chemical control costs for turfgrass professionals. Early diagnosis is vital in turfgrass that is used for sports and recreation, where pathogen damage can severely affect the use of those fields. Updating diagnosis methods in turfgrass has great potential to reduce costs and protect investments within this ever-growing industry. Using new molecular tools can allow for in-field diagnosis and the potential to catch infections before damage. While some of these methods have been utilized in turfgrass, the field is extremely wide on how molecular and advanced pathology detection can grow, with many pathogens that these methods can be used to help control.

In this review, we summarized the current and potential future directions within the turfgrass pathogen diagnosis field. Our review identified that several methods that can effectively help identify pathogens. There are many species-specific primers available for pathogen detection; however, there are still many turfgrass pathogens on conventional PCR technology using universal PCR primers followed by sequencing of the amplified DNA fragments for diagnosis. Such a protocol often takes days for processing and requires pure cultures of a sample—which is often not possible to obtain or might not be present during early stages of infections and/or in the case of latent infection.

There are many directions for turfgrass pathology to grow in the future. Throughout this review several methods were discussed as available, but not commonly used, even the rapid methods that can be used directly in a turfgrass plot. There are several reasons for the disconnect. Arguably the biggest reason is that often a turfgrass manager is going to treat for what they think the disease based on symptoms rather than wait for results of sending the samples to a laboratory or running tests. This immediate action can prevent further damage, and for many cases this will be a good solution. However, if the diagnosis is incorrect this leads to lost time and higher damage. There is also the issue of training and equipment for many of these methods not being widespread within the turfgrass field. To have these tools better utilized it would require training these field managers to use them. There are diagnostic laboratories throughout the United States and the world that these samples can be mailed to for diagnosis, and that is where most of these methods are currently being used. These require shipping time but can help prevent incorrect diagnosis based on symptoms. This can impact the management costs lost to incorrect diagnosis. This offset can be used to cover costs of pathogen identification from these specialized labs.

We hope to see newer methods utilized in the future of the industry and expanding the current methods to identify other turfgrass pathogens. There are many pathogens that could be identified with one of these assays, but that combination has yet to be developed. Expanding available options will help encourage growth in the industry while reducing the damage caused by pathogens. Developing more efficient technologies such as LAMP, RPA, and lateral flow device methods are rapid and highly sensitive for screening early-infected grasses that harbor relatively low pathogen populations. Accurate and rapid detection are crucial for the sustainable disease management by permitting early prediction of disease and reducing the risk of epidemics.

## Figures and Tables

**Figure 2 plants-09-01544-f002:**
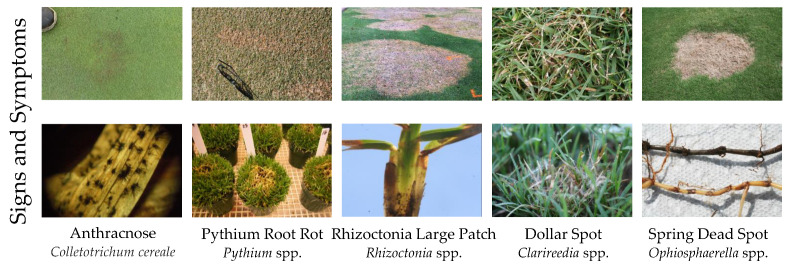
Signs and symptoms of common turfgrass diseases. The signs and symptoms of anthracnose (*Colletotrichum cereale*), pythium root rot (*Pythium* spp.), rhizoctonia large patch (*Rhizoctonia* solani), dollar spot (*Clarireedia* spp.), and spring dead spot (*Ophiosphaerella* spp.).

**Figure 3 plants-09-01544-f003:**
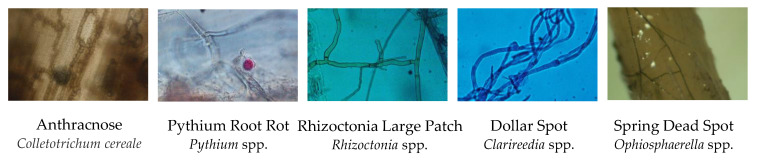
Microcopy of a few common turfgrass diseases. Microscopy photos are shown of anthracnose (*Colletotrichum*
*cereal*), pythium root rot (*Pythium* spp.), rhizoctonia large patch (*Rhizoctonia solani*), dollar spot (*Clarireedia* spp.), and spring dead spot (*Ophiosphaerella* spp.).

**Figure 4 plants-09-01544-f004:**
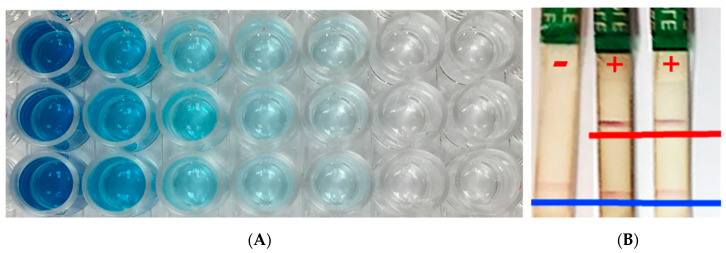
Serological techniques. (**A**) An enzyme-linked immunosorbent assay (ELISA) testing concentrations of a given sample for detection limits. Blue = positive and clear = negative. (**B**) An immunostrip assay with one negative sample and two positives. The blue line indicates the control band and the red line indicates a positive sample band. Photos taken at the Plant Molecular Diagnostic Lab at the University of Georgia, Tifton.

**Figure 5 plants-09-01544-f005:**
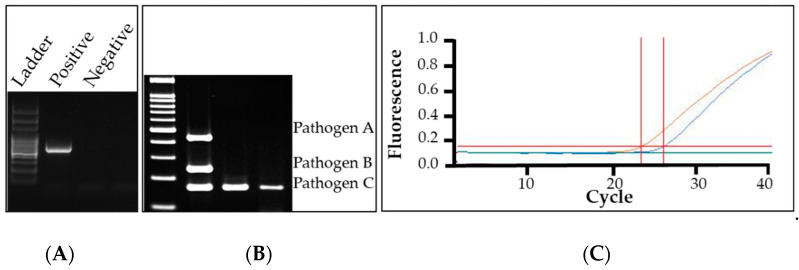
(**A**) Gel electrophoresis results of a conventional PCR reaction with one positive and one negative sample. Photo taken at the Plant Molecular Diagnostic Lab at the University of Georgia, Tifton. (**B**) Results of a multiplex PCR reaction. There are three samples. Sample 1 contained pathogens A, B, and C, sample 2 and 3 contained pathogen C. This figure is modified from Sea-liang et al. (2019) under the Creative Commons Attribution License [77]. (**C**) Results of a qPCR reaction with one positive control (blue), one negative control (green), and one positive sample (orange). The horizontal red line is the threshold that must be passed to be a positive sample. The vertical red line denotes the cycle that the threshold was surpassed.

**Figure 6 plants-09-01544-f006:**
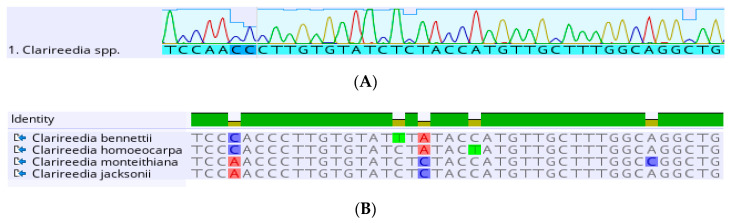
Sanger sequencing pathogen identification. (**A**) Chromatogram of a *Clarireedia* spp. sample, viewing a partial internal transcribed spacer (ITS) region sequence. Each colored peak represents the reading of an individual base with each color representing a different base. Larger peaks are stronger readings. (**B**) Partial sequence comparison of four species within the *Clarireedia* genus. The top line shows how similar the sequences are to one another, while each following line shows the bases of that section of the ITS region. The colored individual bases in the comparison highlight the difference between the individual sequences.

**Figure 7 plants-09-01544-f007:**
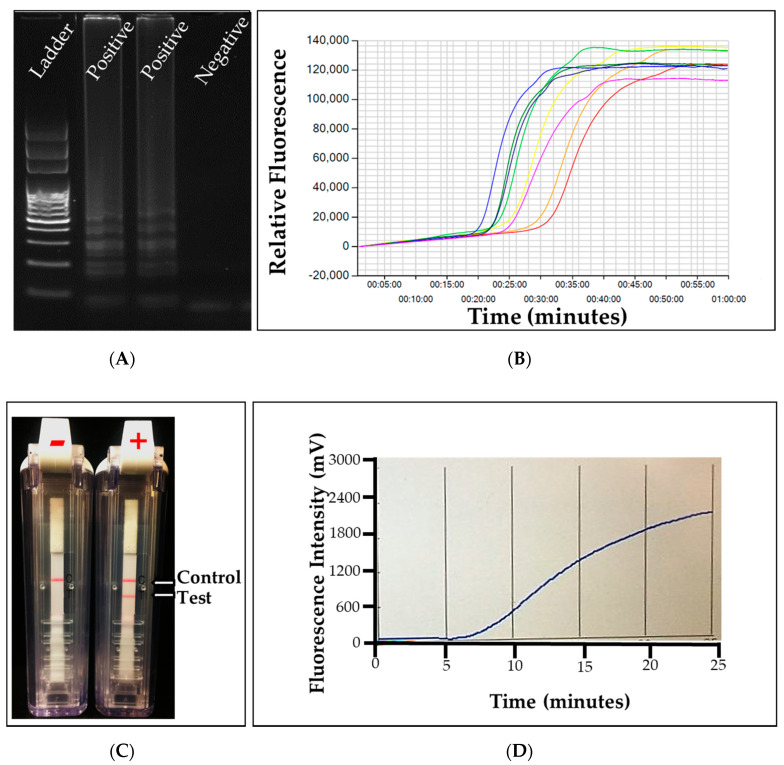
Visualization of isothermal disease detection assays. (**A**) Gel electrophoresis results of a loop-mediated amplification (LAMP) assay with two positive and one negative sample. (**B**) Real time results of a LAMP assay with eight positive samples. (**C**) Recombinase polymerase amplification (RPA) assay visualized with an immunostrip with one negative and one positive sample. The first line indicates the control band and the second line on the second test indicates a positive sample band. (**D**) Real time results of a RPA assay with one positive sample. Images taken at the Plant Molecular Diagnostic Lab at the University of Georgia, Tifton.

**Figure 8 plants-09-01544-f008:**
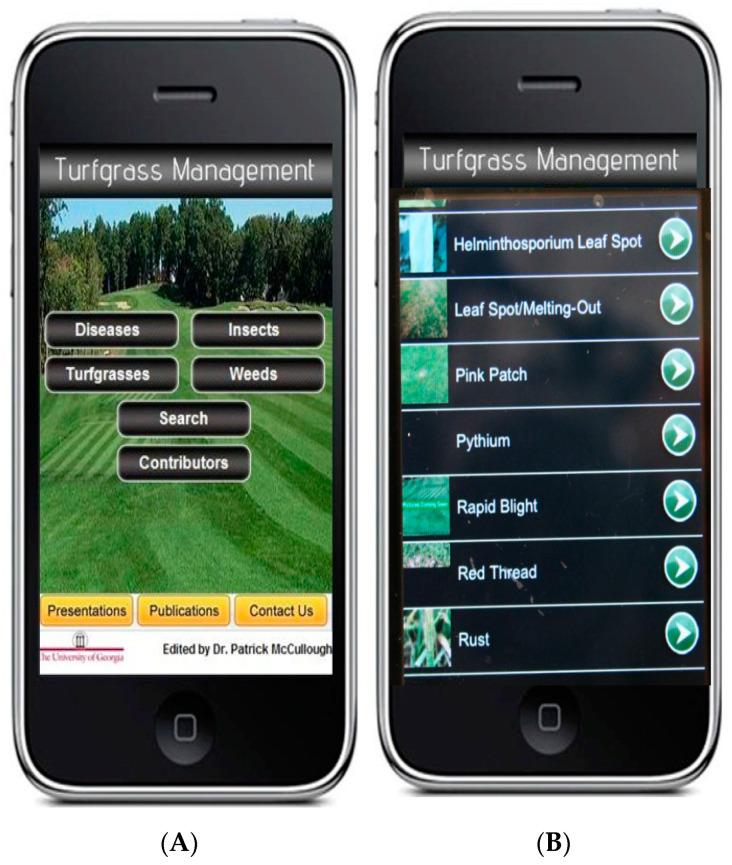
Turfgrass management application. (**A**) Main menu of the first Turfgrass Management™ application put in the play and apple stores in 2005. (**B**) The pathogen details selection section of first turfgrass management application. Selecting a pathogen will give photos and extensive information on that pathogen.

**Table 1 plants-09-01544-t001:** Currently available methods for pathogen detection in turfgrass.

Technique	Common Name	Pathogen Assays	Target Gene/Protein/Technique	Citations
ELISA	Damping-off *	*Pythium* spp.	Commercial *Pythium* antibody	[50]
	Necrotic ring spot	*Leptosphaeria korrae*	(MAb) LKc50 antibody	[51]
	Brown patch	*Rhizoctonia* spp.	Commercial *Rhizoctonia* antibody	[53]
	Damping-off *	*Pythium ultimum*	(MAb E5) antibody	[52]
PCR	Damping-off *	*Pythium* spp.	ITS region	[54]
	Take-all Patch	*Gaeumannomyces graminis* (3 varieties)	ITS; avenacinase-like genes	[43,55,56]
	Brown patch	*Rhizoctonia solani*	RAPD-PCR	[57]
	Fairy ring	*Vascellum pratense*	ITS region	[58]
	Fairy ring	*Lycoperdon pusillum*	ITS region	[58]
	Blast disease	*Magnaporthe oryzae*	Pot2 transposon	[59]
	Dead spot	*Ophiosphaerella agrostis*	ITS region	[60]
	Anthracnose	*Colletotrichum graminicola*	ITS region	[61]
	Necrotic ring spot	*Ophiosphaerella korrae*	ITS region	[62]
qPCR	Dollarspot	*Clarireedia* spp.	ITS region	[63]
	Summer patch	*Magnaporthe poae*	ITS region	[64]
	Brown patch	*Rhizoctonia solani* and *R. oryzae*	ITS region	[65]
	Rust	*Puccinia* spp. (three pathogenic spp.)	ITS region	[66]
	Bacterial etiolation	*Acidovorax avenae*	Draft genome	[67]
Multiplex	Damping-off *	*Pythium* spp. (5 spp.)	ITS region	[54]
PCR	Take-all Patch	*Gaeumannomyces graminis* (3 varieties)	Avenacinase-like genes	[56]
LAMP	Take-all Patch	*Gaeumannomyces avenae*	18S ribosome region	[68]
	Necrotic ring spot	*Ophiosphaerella korrae*	18S ribosome region	[68]
	Summer patch	*Magnaporthiopsis poae*	18S ribosome region	[68]
	Gray leaf spot	*Magnaporthe oryzae*	Draft genome	[69]
	Brown patch	*Rhizoctonia solani*	ITS region	[70]
	Bacterial wilt	*Xanthomonas translucens*	Draft genome	[71]
	Root-Knot Nematodes	*Meloidogyne chitwoodi* *Meloidogyne fallax*	18S ribosome region	[72,73]
RPA	Take-all Patch	*Gaeumannomyces avenae*	18S ribosome region	[68]
	Necrotic ring spot	*Ophiosphaerella korrae*	18S ribosome region	[68]
	Summer patch	*Magnaporthiopsis poae*	18S ribosome region	[68]

* This pathogen has several common names.

**Table 2 plants-09-01544-t002:** Advantages and disadvantages of pathogen detection methods.

Diagnostic Technique	Method Advantages	Method Disadvantages
Morphology, signs, and symptoms	There are very little inputs, and this method is extremely rapid.	With many diseases it is incredibly difficult to identify with morphology. It requires extensive knowledge. Many diseases have similar symptoms. Pathogens can also have similar signs.
Culturing and microscopy	These methods can be very rapid tests for some species with little equipment needed.	This method requires extensive knowledge and may not be able to identify down to species. Many pathogens are not culturable.
ELISA	ELISA does not require a DNA extraction and is commercially available. Quantifiable with a plate reader and controls and very cheap per sample. Visual results allow for simple qualitative interpretation.	This method takes about 24 h to run, requires a laboratory, is not as sensitive as other methods, and has a lot of hands on steps.
PCR	PCR has both universal primers for unknown samples and species-specific primers. Easy to run with commonly available equipment.	PCR is not as sensitive as qPCR, RPA, or LAMP and typically requires a lab setting and expertise. It sometimes needs an extraction, which can take 1–4 h.
qPCR	qPCR is an extremely sensitive method and has quantification. It is also faster than conventional PCR or ELISA.	qPCR requires a lab setting and very expensive equipment and moderately expensive reagents. Contamination can be common.
Multiplex PCR	Multiplex PCR allows for several tests to be performed at one time in the same assay. The ability to run several tests at once makes multiplex PCR have fewer inputs than conventional PCR.	Multiplex requires a lab setting and primer design can limit use of this method.
LAMP	LAMP is an isotheral reaction, which allows for in-field diagnosis.	LAMP is prone to false positives from contamination and similar issues.
RPA	RPA is an extremely sensitive isothermal reaction.	RPA is difficult to design primers and may be less sensitive to mismatches. This makes species-specificity more difficult.
Lateral flow assays	Lateral flow assays are field ready and are extremely rapid tests. They are simple will very little training needed to run the assay.	Lateral flow assays have lower sensitivity than any of the nucleic acid-based assays.
Portable high-throughput sequencing	High-throughput sequencing allows for testing without any information on the pathogen. It is sensitive and has mixed sample capability.	Portable high-throughput sequencing is still extremely expensive comparatively to all other methods. It produces a lot more data than is typically needed and often requires sorting through nonpathogenic results.
Remote Sensing	Remote sensing allows for detection before visual symptoms are present	Remote sensing is expensive and limited in its current data. It requires expensive equipment and human input in data analysis.

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
