# Peer review of "Turfgrass Disease Diagnosis: Past, Present, and Future"

_plants, 2020, doi:10.3390/plants9111544_

Round 1
Reviewer 1 Report
Review of Turfgrass Disease Diagnosis: Past, Present, and Future
Stackhouse et al. 2020
This review is a useful discussion of techniques. However, serious thought needs to occur regarding the difference between diagnosis of disease and identification of organisms. Additionally, discussion in needed regarding the utility in a practical sense for disease management, or application in a service program. Discussion of pros and cons of various methods is uneven, as is the depth of explanation of each method’s technology. Specific edits follow, though grammar editing will be necessary above and beyond what is noted here.
Line 41 add “be” (“be confused”)
Line 47: remove the extra “and”
Line 63: if an organism is not causing symptoms, is it causing disease? Does it need to be managed? Many organisms can be apparent on the host, but not causing damage or disease unless other parts of the disease triangle are in effect – conducive environment, susceptible host, etc. Pathogen detection and disease diagnosis are not the same thing.
Line 80 (timeline): Nice graphic. Check text in the graphic for capitalization consistency. Since this is a review paper, I think you should have citations to back up the timeline dates and to give readers quick access to the techniques if they need more information. Immunostrip is a specific proprietary product. The generic is called a lateral flow device. Also, there is a difference between genome sequencing and whole-genome sequencing and high-throughput sequencing; consider the addition of these. minION is specific to Nanopore; consider revising to state as generic. Also, confirm minION release date (2014?).
Line 88: pathogens have signs; diseases have symptoms. Use consistent terminology. Recommend introducing the concept of diagnosis as a process of synthesis – symptoms, identification of organisms, determination of disease versus presence of an organism, association with cultural or environmental issues conducive to disease development.
Line 104, Culturing and Microscopy section: one diagnoses diseases and identifies pathogens.
Images are nice, but these images may not be the best and the use of micrographs vs culture is not consistent – only one culture image is notes, the others are all micrographs: Pythium images don’t appear to show characteristic sporangia or oospores; the two images of Rhizoctonia are showing the same thing – why include both – maybe a culture image would be more useful; the bottom right micrograph looks more like Gaeumanomyces than Ophiosphaerella, given the prominent hyphapodia. What is the point of this figure? To illustrate the difficulty of diagnosis by microscopy?
Line 130: DAS is used for several reasons, but ELISA is not generally considered quantitative unless you have validated controls and use a plate reader; even then this is really a qualitative test, as you note in line 134 – positive or negative. Some more discussion about specificity is needed. Line 136 – change While to With? Line 138 – although there are some antisera that are useful for detecting some turf pathogens, I disagree that ELISA is widely used in the turf industry. Current use of the SCMV DAS ELISA from Agdia is used in Florida diagnostic labs on St. Augustinegrass, and some labs might use a Pythium test for root rots and blights, but to say the industry regularly uses ELISA is misleading unless you have data to back this up. For this reason, Table 1 should be retitled to indicate potential serological and molecular tools for turf pathogen identification – these are not methods commonly used presently, as you indicate about immunostrips in the section 3.2.
Line 149: the generic term for an immunostrip method is lateral flow device.
Line 152 – antibodies do not search for antigens; they diffuse into contact with them, hence the need for incubation periods in the process. And in lateral flow, capillary action carries antibodies into contact with the antisera. Consider a real rewrite of this section.
Line 163: Nucleic Assay Techniques – these are not techniques to study nuclei, they are assays that target nucleic acids. Retitle this section.
Line 175: this is probably how the priers SHOULD be designed, but are they really tested in silico against “ALL the sequences in GenBank”? Please rewrite this section to reflect protocol development challenges – preliminary diagnosis required (known target) and challenge of developing robust primers that differentiate well. This whole paragraph skips around from defining the process of PCR to challenges and benefits. Additionally, genus-specific diagnosis may have been available for over two decades, but why it hasn’t been widely applied should be discussed here. Interpretation of results should be discussed.
Line 183: Figure 5 A should be better explained in the conventional PCR paragraph to indicate how positive and negative should be interpreted. Negative is not only the absence of a band, but also the presence of bands in the wrong/unexpected size. 5B: Additionally, a gel showing positive and negative control lanes, as well as ladder and samples, is the ideal image. 5C: positive control should be included as well.
Line 190: “threshold that must be passed”, not past.
Line 196: “read on a gel electrophoresis”? The gel image description should be in the paragraph about conventional PCR, then the paragraph about multiplex should just indicate what is different about multiplexing, since you can multiplex more than conventional PCRs. Also, explain the drawbacks/challenges as well as the benefits (why is it nice to have them multiplexed?)
Line 198: Pythium spp., not Pythium sps.
Line 206: explain why this is quantitative. This is an important benefit of qPCR.
Line 208: fix this sentence fragment with they stray question mark. Also, human pathogen detection, not disease detection.
Line 212: relatively slow process compared to what? You don’t mention time in the conventional or multiplex PCR sections.
Line 223: revise sentence fragment.
Line 229: I think you mean all you have to know is it is a bacterium so you can choose appropriate “universal” primers? Also, please discuss challenges to making this work – the need for pure cultures and clean amplicons to sequence. Also, deciding which primers to use, specifically, requires more than knowing a bit about the organism, oomycete vs true fungi, for instance, or family of virus. Additionally, whether the virus has a coat protein, or requires an RT step prior to amplification (RNA vs DNA viruses).
Line 242: Figure 6: ID of a pathogen is not disease diagnosis; please revise figure title.
Line 282: Does the turf industry have many or a few LAMP assays? These two sentences seem to contradict each other. Or do you mean there are assays available, but few are actually in use in the industry? Please discuss challenges with LAMP – why isn’t this being used more?
Line 295: Figure 7D: there is only one blue line, but the caption mentions seven others?
Line 323: remove the word “control” – the paper is about diagnosis/detection.
Line 325 – consider revising “diagnosis” to “detection” for accuracy, and remove comma after diagnosis. Remove d in “remoted”.
Line 338: what are the limitations/challenges?
Line 353: What are the drawbacks? Why isn’t it being used more? Why the use of the word infestation when this is about disease?
Line 355: Table 2: revise title to pathogen detection. Table is very useful. Edit for grammar throughout.
Line 370: Change pathogens to diseases
Line 378: Edit for capitalization, trademarks, etc.
Conclusions: Consider discussing how advanced techniques impact disease management and whether speed or specificity gains can offset the costs to the lab or the client.
Author Response
Dear Reviewer,
Please see the attached file for our responses to your questions.
Thank you,
Ali

Reviewer 2 Report
Review paper should present common and international interest on the subject not local or regional interest unless this was indicated in the title. The title of the review is not a good fit with the content. The title focusses on the past, present and the future of Turfgrass diseases in general while the review focusses on specific region or specific types of turfgrasses. Have you discussed cold season turfgrasses in details? Have you reviewed the work done in the Midwest or in the North.
Also, requires some extensive editing and review. Reference list needs to be well reviewed. For example, first reference on the list is not stated in a correct way.
Chawla, S.; Roshni, A.; Patel, M.; Patil, S.; Shah, H. Turfgrass: a billion dollar industry. In Proceedings of
National Conference on Floriculture for Rural and Urban Prosperity in the Scenerio of Climate Change-
2018.
Can anyone get access to this reference or know where it was held or where it was published. What is the importance of listing something like that? It is not professional. I searched and found some more information is needed for the publisher, number of pages, ……etc.
Additional comments include:
Line: 24: delete repeated keywords in manuscript title
Missing DOI of some references must be add
Publication year must be bold
Line: 437 Journal title is missing (HortScience)
Line 442: publication year is missing
Line 468, 469: correct, there is two year for this reference (2017 or 2019)
Line 478: journal name (first letter must be Caps)
Line 481, 482, 483: add the type of these publications
Line 499: publication year is missing
Line 540: add volume, number and page numbers of the journal
Line 572: journal name (first letter must be Caps)
Line 577: publication year is missing
Line 631: add volume, number and page numbers of the journal
Line 634: add volume, number and page numbers of the journal
Line 694: add all authors’ name
Line 752: journal name (first letter must be Caps)
Line 757: publication year is missing
Author Response
Dear Reviewer,
Please see the attached file for our responses to your comments.
Thank you,
Ali

Round 2
Reviewer 2 Report
The manuscript is significantly improve. Double check the spelling and language problems befor publication, and double check all scientific naes.